# Blacks’ Diminished Return of Education Attainment on Subjective Health; Mediating Effect of Income

**DOI:** 10.3390/brainsci8090176

**Published:** 2018-09-12

**Authors:** Shervin Assari

**Affiliations:** 1Center for Research on Ethnicity, Culture, and Health (CRECH), School of Public Health, University of Michigan, Ann Arbor, MI 48104, USA; assari@umich.edu; Tel.: +1-734-363-2678; 2Department of Psychiatry, University of Michigan, 4250 Plymouth Rd.; Ann Arbor, MI 48109-2700, USA

**Keywords:** population groups, race, ethnicity, African Americans, Blacks, socioeconomic status, self-rated health, economic inequalities, ethnic health disparities

## Abstract

**Background:** Minorities’ Diminished Return (MDR) can be defined as smaller health gains from socioeconomic status (SES) indicators, such as education attainment among ethnic minorities compared to the majority group. The current study tested whether income explains why Black and White adults differ in the association between education attainment and self-rated health (SRH). **Methods:** With a cross-sectional design, this study used data from Cycle 5 of the Health Information National Trends Survey (HINTS), 2017. With a nationally representative sample, the HINTS study generates results that are generalizable to US adults. This study included 2277 adults who were either non-Hispanic White (*n* = 1868; 82%) or non-Hispanic Black (*n* = 409; 18%). The independent variable was education attainment. The dependent variable was SRH, measured using a standard single item. Age, gender, and health insurance status were covariates. Ethnicity was the focal moderator. Income was the mediator. A structural equation model (SEM) was applied for data analysis. **Results:** Overall, higher education attainment was associated with better SRH, net of covariates. However, a significant interaction between ethnicity and education attainment suggested a smaller SRH gain from education for Blacks compared to Whites. This interaction could be explained by Black–White differences in income. **Conclusion:** Our study results suggests that labor market preferences may explain smaller effects of education attainment on SRH for Blacks relative to Whites. Given this finding and other studies documenting MDR, policies should reduce labor market discrimination, increasing job opportunities and reducing the racial pay gap for Blacks. Programs should help Blacks compete for prestigious and high-paying jobs.

## 1. Introduction

An overwhelming body of literature suggests that high socioeconomic status (SES) promotes population health [1,2,3]. The health effects of SES indicators such as education attainment [4] and income [5], against risk of morbidity [6] and mortality [1,2] are well established. The protective effect of high education attainment against poor health is well documented by original [7,8,9,10,11,12,13,14,15] and review [16] studies. High education is shown to protect against a decline in self-rated health (SRH) [17,18,19], chronic disease [20], and mortality [21].

According to the Minorities’ Diminished Return (MDR) theory [22,23,24,25,26], ethnic minorities gain disproportionately less health benefits from their SES resources compared to White Americans [27,28,29,30,31,32,33,34,35,36]. Black Americans may gain less health in comparison to White Americans from a wide range of SES resources [30,31,36], including education [28] and income [25,28,37].

The mechanism behind the diminished return of education for Blacks compared to Whites is still unknown. In recent studies, discrimination has been proposed as a potential mechanism [38,39]. However, the results on the effects of discrimination are mixed [40,41]. Despite equal resources and social capital across ethnic groups, racism and discrimination may limit the health gains that follow SES for ethnic minorities dealing with a wide range of barriers in their daily lives [30,31]. Other structural and societal factors, such as residential segregation and discrimination by labor markets, contribute to the relative disadvantage of Blacks compared to Whites in terms of diminished gains from education [30,31], partly because education generates less economic return for Blacks [23,34,42,43].

Still less is known about the exact mechanism behind differential effects of education attainment on SRH across various ethnic groups. One study showed that similar education attainment results in lower income for Blacks compared to Whites [42]. In one study, baseline education predicted future increase in income for Whites, but not Blacks [23]. These studies propose that differential income may be one reason why equal education results in unequal health gains for Blacks and Whites. 

The aim of the current study was to compare Blacks and Whites for the association between high education attainment and SRH. Guided by recent literature on disproportionately smaller health gains of non-Whites than Whites from SES indicators [32,44,45], we expected a weaker association between education attainment and SRH for Blacks in comparison to Whites. We hypothesized that income may have some role in explaining the differential gains from education attainment.

## 2. Methods

### 2.1. Design and Setting

The current cross-sectional study used data from Cycle 5 of the nationally representative Health Information National Trends Survey (HINTS), 2017. HINTS 5, Cycle 1, was conducted from January until May 2017. Supported by the National Cancer Institute (NCI), HINTS provide a comprehensive assessment of cancer information in American adults [46,47].

### 2.2. Ethics

The HINTS 5 study protocol was approved by the Westat’s Institutional Review Board (IRB) but was deemed exempt from IRB review by the NIH (National Institutes of Health) Office of Human Subjects. All HINTS participants provided informed consent. Monetary incentives were included in survey envelopes to encourage participation.

### 2.3. Sampling 

The HINTS target population is American adults (age ≥ 18) who reside in the US and are non-institutionalized. The HINTS 5, Cycle 1 used a two-stage sampling design. First, a stratified sample of addresses was derived from all residential addresses. In the second stage, one adult was drawn from each sampled household. The address list was provided by the Marketing Systems Group (MSG), and included all non-vacant residential addresses in the US. The sampling frame was composed of the following two sampling strata: (1) high minorities concentration areas; and (2) low minorities concentration areas. An equal-probability sample was drawn from each of the two strata [46,47].

### 2.4. Surveys

The HINS survey was conducted exclusively by mail. Two toll-free telephone numbers were provided for participants to participate in the survey: one line for English calls and one line for Spanish calls [46,47].

### 2.5. Measures

Study variables included age, gender, health insurance, ethnicity, depression, body mass index, smoking, exercise, household income, and SRH.

#### 2.5.1. Independent Variables

*Education Attainment* was the main independent variable in our study. Educational level was measured as a five-level ordinal variable: (1) less than high school; (2) high school graduate; (3) some college; (4) bachelor’s degree; and (5) post-baccalaureate degree. In this study, education attainment was operationalized as a continuous measure, with a higher score reflecting more education attainment.

#### 2.5.2. Dependent Variable

*Self-Rated Health* (*SRH*) was measured using a single item. Participants were asked about their overall health. Responses included excellent, very good, good, fair, or poor. The literature has mostly treated SRH as a dichotomous variable, combining poor and fair categories compared to any other responses [48,49,50]. Poor/fair SRH was coded as 1 [51]. The Institute of Medicine (IOM) has recommended the use of SRH for monitoring the health of Americans [49]. SRH has high validity, as it predicts long-term risk of mortality beyond health-related covariates [52,53,54,55]. 

#### 2.5.3. Covariates

Age, gender, and health insurance were the study covariates. Age was a continuous measure. Gender was a dichotomous variable (men 0 (reference group) and women 1). Health insurance was treated as a dichotomous variable (0 without insurance, 1 with insurance). We considered the following types of insurance: (1) Insurance purchased directly from an insurance company; (2) Medicare, for people 65 and older, or people with certain disabilities; (3) Medicaid, Medical Assistance, or any kind of government-assistance plan; (4) TRICARE or other military health care; (5) Veteran Affairs (including those who have ever used or enrolled for VA health care); (6) Any other type of health insurance or health coverage plan. 

#### 2.5.4. Moderator

Ethnicity. Self-identified ethnicity was the focal moderator. Ethnicity was operationalized as a dichotomous variable (Whites 0 (reference group), Blacks 1).

#### 2.5.5. Mediator

*Household Income.* Household income was treated as a continuous measure. Income was measured as (1) $0–9999, (2) $10,000–14,999, (3) $15,000–19,999, (4) $20,000–34,999, (5) $35,000–49,999, (6) $50,000–74,999, (7) $75,000–99,999, (8) $100,000–199,999, and (9) $200,000 or more. 

### 2.6. Statistical Analysis

*Sampling and Design Weights.* We used Stata 15.0 (Stata Corp., College Station, TX, USA) for our data analysis to accommodate the HINTS multi-stage sampling design. Jackknife was used to re-estimate standard errors based on weights due to strata, clusters, and non-response. All analyses were run using sub-population survey commands. 

*Analytical plan.* For univariate statistics, we provided mean and proportions (frequencies) to describe the distribution of our variables in the pooled sample as well as by ethnicity. For multivariable analysis, we used structural equation modeling (SEM) [56] to test the effects of education attainment on health. We ran the following seven models: *Model 1* (main effect model without any mediator), *Model 2* (interaction model without any mediator), *Model 3* used income as the mediator. We reported path coefficients, SE (Standard Errors), 95% CI (Confidence Intervals), *z* value, and *p* value. *p* < 0.05 was considered significant.

To handle missing data in Stata, we used maximum likelihood estimates [57,58]. Fit was considered to be acceptable with a chi-square to degrees of freedom ratio of less than 4, a comparative fit index (CFI) above 0.95, and a root mean square error of approximation (RMSEA) value of 0.06 or less [59,60].

## 3. Results

### 3.1. Descriptive Statistics

This study included 2277 adults who were either non-Hispanic White (*n* = 1868; 82%) or non-Hispanic Black (*n* = 409; 18%). Non-Hispanic Blacks had lower education attainment and household income than non-Hispanic Whites. Non-Hispanic Blacks also reported worse SRH than non-Hispanic Whites.

### 3.2. Multivariable Models

SEMs showed acceptable fit. Table 1 provides a summary of seven SEMs. Based on *Model 1*, high education attainment was associated with better health, independent of covariates. Based on *Model 2*, there was a significant interaction between ethnicity and education on health, suggesting that Blacks gain less health from their education attainment compared to Whites. 

Based on *Model 3*, income explained the differential effect of education by race/ethnicity on SRH. That is, after adding income to the model as a potential mediator, the effect of interaction term between race/ethnicity and education on health was no longer significant (Figure 1, Figure 2 and Figure 3).

## 4. Discussion

The current study showed an ethnic variation in the association between education attainment and poor SRH, and household income may explain the differential gain of education attainment by ethnicity. In simple terms, income may be one of the main reasons why education attainment is associated with better health for Whites than Blacks. 

This study suggests that there might be economic and behavioral explanations for the diminished return of education for Blacks. Our first finding showed that lower household income and behavioral factors of highly educated Blacks may explain why Blacks gain less SRH than Whites from the same education attainment. As income is involved, labor market discrimination, racism, and segregation shape how competitive Blacks are compared to Whites to secure high paying jobs. Such differences may also impact behaviors such as smoking and obesity. 

While education and income impact a wide range of health outcomes, including but not limited to SRH, this finding is anything but new. Fundamental cause [61,62] and social determinants of health [63,64] theories have thoroughly explained how SES and other resources impact health. This finding is also supported by many cross-sectional studies and cohorts with long follow-up periods, suggesting that high SES impacts SRH [17,18,19]. SES operates via various mechanisms, including but not limited to mental health [16] and health behaviors [65,66,67,68,69]. The protective effects of SES on health are non-specific, as the health benefits of SES resources extend to multiple outcomes [70]. 

This is not the first study to show that Blacks are at a disadvantage in translating their economic resources to health outcomes. Similar effects are shown for physical and mental health outcomes [71,72,73,74,75]. The unique contribution of this study is to show that household income may have some role in explaining such ethnic variations. Such a finding proposes racial pay gap as a mechanism for findings shown by the MDR theory [30,31].

Education and income better promote self-rated health [22,76], self-rated mental health [77], and self-rated oral health [78] for Whites than Blacks. Income may be a mechanism for why education generates more health for Whites than Blacks. Household income better reduces chronic medical conditions for Whites than Blacks [32]. Differential effects of education on income or stronger effect of income on health of Whites may be due to structural factors such as residential segregation, which reduces Blacks’ access to resources, job availability, and healthy food choices, even in the presence of income. The same income generates lower purchasing power, given Blacks’ tax and the recency of Blacks to their social class [79].

### Limitations

This study is not without limitations. First, HINTS sampling was limited to individuals with phones. Second, with a cross-sectional study, we are unable to infer causal associations. SES and health have reciprocal associations. Further, low SES and downward social mobility reduce health and upward social mobility enhances health. At the same time, good health helps people with upward social mobility maintain their SES [80]. Future research should conceptualize SES as time-varying and should study changes in SES and health over individuals’ life course. This is particularly important, as social mobility may have differential effects on the health of Whites and Blacks [81]. Another major limitation of the study was residual confounding. This study did not include potential confounding factors such higher level SES, access to health care system, and health care use. We also did not include other SES indicators such as marital status, wealth, employment, and occupation. Our outcome (SRH) was also a single-item measure. Future research should extend the results of this study by studying other measures such as psychiatric disorders, chronic disease, and mortality. The differential validity of SRH by ethnicity is a threat to the validity of current study. SRH may have different meanings across ethnic groups, and poor SRH may reflect different domains of health and well-being for Blacks compared to Whites [82]. While education was an individual characteristic, income was household income. Future research should include data from individual, family, community, and neighborhood levels. Contextual factors such as density of poverty, ethnic groups, jobs, resources, community safety, public transport, and other measures may explain why Whites and Blacks do not similarly gain health from the same resources [83,84,85]. Future research should replicate these findings in independent data sets—particularly various designs, settings, cohorts, and age groups. 

## 5. Conclusions

Higher education attainment was linked to better self-rated health. The magnitude of this association was a function of ethnicity, and income may have a role in explaining these differential effects by race and ethnicity. Additional research is needed to better understand the policies and programs that can narrow Blacks’ diminished returns from education attainment. 

## Figures and Tables

**Figure 1 brainsci-08-00176-f001:**
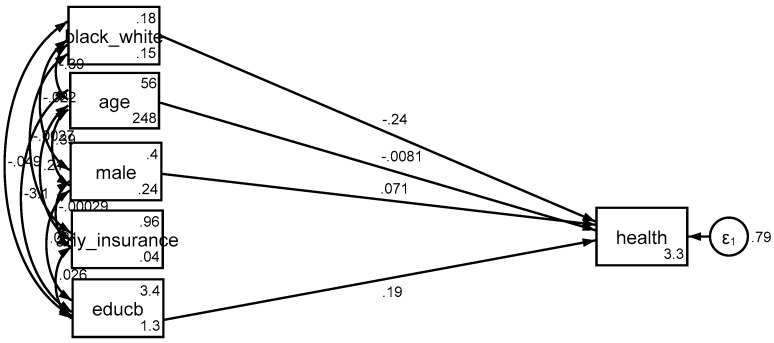
Summary of the results of *Model 1* (main effect model without any mediator).

**Figure 2 brainsci-08-00176-f002:**
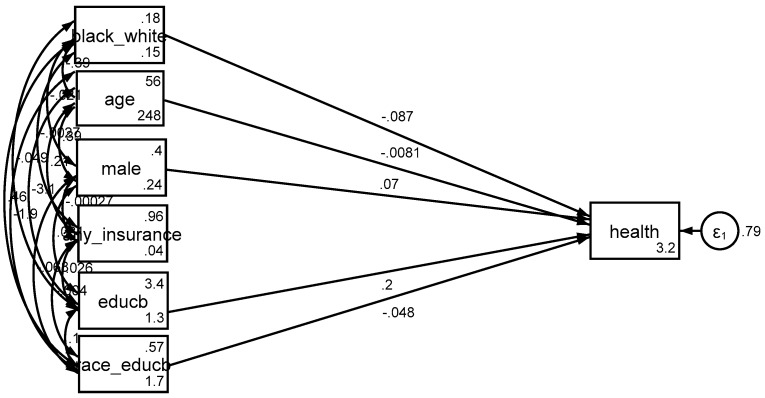
Summary of the results of *Model 2* (interaction model without any mediator).

**Figure 3 brainsci-08-00176-f003:**
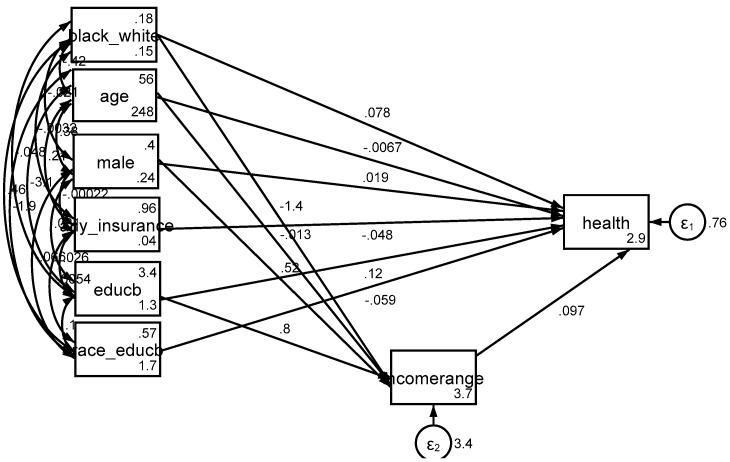
Summary of the results of *Model 3* (income as a mediator).

**Table 1 brainsci-08-00176-t001:** Summary of structural equation models (SEMs).

		b	SE	95% CI		*z*	*p*
***Model 1 (Main Effect Model)***							
Age	Health	−0.01	0.00	−0.01	−0.01	−7.00	0.000
Gender (Male)	Health	0.07	0.04	0.00	0.14	1.93	0.053
Ethnicity (Blacks)	Health	−0.24	0.05	−0.34	−0.14	−4.87	0.000
Education	Health	0.19	0.02	0.16	0.22	11.66	0.000
Intercept	Health	3.26	0.10	3.07	3.44	33.79	0.000
***Model 2 (Interaction Model)***							
Age	Health	−0.01	0.00	−0.01	−0.01	−7.00	0.000
Gender (Male)	Health	0.07	0.04	0.00	0.14	1.90	0.057
Ethnicity (Blacks)	Health	−0.09	0.14	−0.37	0.19	−0.61	0.543
Education	Health	0.20	0.02	0.16	0.23	10.86	0.000
Ethnicity (Blacks) × Education	Health	−0.05	0.04	−0.13	0.03	−1.15	0.249
Intercept	Health	3.22	0.10	3.03	3.42	32.01	0.000
***Model 7 (Income)***							
Income	Health	0.10	0.01	0.08	0.12	9.92	0.000
Age	Health	−0.01	0.00	−0.01	0.00	−5.84	0.000
Gender (Male)	Health	0.02	0.04	−0.05	0.09	0.53	0.596
Health insurance	Health	−0.05	0.09	−0.22	0.12	−0.55	0.583
Ethnicity (Blacks)	Health	0.08	0.14	−0.20	0.36	0.55	0.583
Education	Health	0.12	0.02	0.08	0.16	6.34	0.000
Ethnicity (Blacks) × Education	Health	−0.06	0.04	−0.14	0.02	−1.42	0.155
Intercept	Health	2.90	0.13	2.65	3.14	23.09	0.000
Age	Income	−0.01	0.00	−0.02	−0.01	−5.34	0.000
Gender (Male)	Income	0.52	0.08	0.36	0.67	6.55	0.000
Ethnicity (Blacks)	Income	−1.39	0.11	−1.60	−1.18	−13.08	0.000
Education	Income	0.80	0.03	0.73	0.87	23.02	0.000
Intercept	Income	3.70	0.21	3.29	4.11	17.76	0.000

Notes: Source: Cycle 5 of the Health Information National Trends Survey (HINTS), 2017. b: Unstandardized regression Coefficient; SE: Standard Error; CI: Confidence Interval.

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
