# Peer review of "Blacks’ Diminished Return of Education Attainment on Subjective Health; Mediating Effect of Income"

_brainsci, 2018, doi:10.3390/brainsci8090176_

Reviewer 1 Report
This study examines the important issue of the diminishing returns hypothesis in an effort to identify factors that explain why Blacks seem to gain fewer health returns for higher education.
While this is a great topic, there are several issues that currently limit the paper's overall contribution.
Substantial English language editing is needed to improve the clarity and grammar of the writing.
The first two paragraphs of the introduction section are repetitive and could be streamlined with the argument: (a) higher SES has been associated with more positive outcomes in the general population, (b) education is an indicator of SES [briefly explain the mechanisms linking it to positive health, (c) diminishing returns studies show that Blacks receive fewer health benefits with higher education levels.
The introduction would be strengthened with more focus on the mediating factors examined. Since this is the focus of the paper, elaborating on why these factors are potentially important and should be studied would strengthen the authors' arguments. Tell us why this is an important question. The authors could possibly reduce the content of the third paragraph (lines 47-57) since this is not central to the main focus of the paper.
In the measurement description, only income is described as a mediator although there are others tested. These should be added to the description as well.
The authors should consider another method by which to report their results. There are quite a number of tables, so it would be helpful to condense this information somehow.
Given the disparate numbers of Blacks versus Whites in the data, was this the best data source to address this question? Many studies examining Black-White disparities have oversampled low-SES Blacks, contributing to the conflation of race vs. class effects. Does the present data set include a socioeconomically diverse sample of Blacks to avoid this issue?
Author Response
We sent our paper for English language edit. I hope this has improved the clarity and grammar of the writing.
We made some modifications to the first two paragraphs of the introduction section, to avoid the repetitiveness. As the reviewer suggested, now are saying: 1) higher SES has been associated with more positive outcomes in the general population, 2) education (and income) are indicators of SES, and 3) diminishing returns studies show that Blacks receive fewer health benefits with higher education levels.
OK. Now we are only focusing on income as our mediator. As a result, the introduction has added a paragraph on income as a mediator. This has hopefully strengthened the introduction. The introduction now elaborates on whisfactors are potentially important and should be studied .
Now, we only have income as a mediator. So, in the measurement description, only income is described as a mediator.
Only focusing on income as a mediator help us have less tables, and more readable paper. So, the results are now condensed.
This study was used for a reason. We did not need low educatyed and low income Blacks. We needed mofderate income Black, because we are arguing that high education is still not protective for Blacks. We still proposed that there is a need to replicate these findings in other data sets.
Reviewer 2 Report
This certainly is an important topic with high interest to readers.
You must justify the scientific reason why your five mediating variables might matter to explain education-SHR covariance beyond the fact HINTS happens to ask about them. You also need to differentiate between each of these five variables and their effects as mediators rather than saying "they all matter" without any differentiation. In general, more detailed and thorough explanation of your methods and findings would greatly benefit the paper.
Significant English-language editing work is required.
The Introduction feels like an ad hoc laundry list of "things that might matter" concerning the relationship between race/ethnicity and health. This needs to be significantly tightened up to focus on your specific data, the mechanisms suggested by your SEM models, and existing literature. It's all over the place in its current form.
Author Response
The main concern was about the kitchen sink / laundry room approach, that I had tested any thing that was a part of the study and could potentially mediate differential effects of education by race. However, as the reviewer suggests, this was more due to the data availability not theory.
In response to the critic, I decided to limit the purpose of the study to economic mechanism, which is both supported by theory, and also previous research.
I added a paragraph about income as a mechanism for differential effect of education to the introduction.
So the revised paper is not all over the place, as I have focused on the differential effects of education and income and then on income as a potential mediator.
The new title also suggests that income is the mediator of interest.
I have also deleted irrelevant data from results, and methods. Only income is left. BMI, depression, smoking, and exercise are left out.